First report of ‘Candidatus Phytoplasma asteris’ associated with yellowing, scorching and decline of almond trees in India

Gupta Shivani 1
Handa Anil anilhanda@msn.com 1
Brakta Ajay 1
Negi Gulshan 1
Tiwari Rahul Kumar 2
Lal Milan Kumar 2
Kumar Ravinder chauhanravinder97@gmail.com 2
1 Plant Virology Laboratory, Department of Plant Pathology, College of Horticulture, Dr. YS Parmar University of Horticulture and Forestry , Solan , Himachal Pradesh , India
2 ICAR-Central Potato Research Institute , Shimla , Himachal Pradesh , India
Shah Anis
Electronic publication date: 2023 Aug 28
Publication date: 2023
Volume: 11
Electronic Location ID: e15926
Received 2023 Jun 1; Accepted 2023 Jul 30
Copyright: ©2023 Gupta et al.
Copyright year: 2023
Copyright holder: Gupta et al.
License: This is an open access article distributed under the terms of the Creative Commons Attribution License, which permits unrestricted use, distribution, reproduction and adaptation in any medium and for any purpose provided that it is properly attributed. For attribution, the original author(s), title, publication source (PeerJ) and either DOI or URL of the article must be cited.
License URL: https://creativecommons.org/licenses/by/4.0/

Keywords: Decline, Fluorescent microscopy, Inward rolling, Nested PCR assays, Reddening, Phytoplasma, Prunus dulcis, Phylogenetic analysis

Funding: The authors received no funding for this work.

==============================
The almond, a commercially important tree nut crop worldwide, is native to the Mediterranean region. Stone fruit trees are affected by at least 14 ‘Candidatus Phytoplasma’ species globally, among which ‘Candidatus Phytoplasma asteris’ is one of the most widespread phytoplasma infecting Prunus dulcis, causing aster yellows disease. Recently, almond plantations of Nauni region were consistently affected by phytoplasma, as evidenced by visible symptoms, fluorescent microscopic studies and molecular characterization. During several surveys from May to September 2020–2022, almond aster yellows phytoplasma disease showing symptoms such as chlorosis, inward rolling, reddening, scorching and decline with an incidence as high as 40%. Leaf samples were collected from symptomatic almond trees and the presence of phytoplasma was confirmed through fluorescent microscopic studies by employing DAPI (4, 6-diamino-2-phenylindole) that showed distinctive light blue flourescent phytoplasma bodies in phloem sieve tube elements. The presence of phytoplasma in symptomatic almond trees was further confirmed using nested PCR with specific primer pairs followed by amplification of 16S rDNA and 16S-23S rDNA intergenic spacer (IS) fragments. Sequencing and BLAST analysis of expected amplicon of the 16S rDNA gene confirmed that the almond phytoplasma in Himachal Pradesh was identical to the aster yellows group phytoplasma. Phylogenetic analysis of 16S rDNA almond phytoplasma also grouped ‘Prunus dulcis’ aster yellows phytoplasma within 16SrI-B subgroup showed 94% nucleotide identity with ‘Prunus dulcis’ phytoplasma PAEs3 and ‘Prunus dulcis’ phytoplasma PAE28 from Iran. This research presents the first host report of ‘Candidatus Phytoplasma asteris’ infecting almonds in India, expanding the knowledge of the diversity and distribution of phytoplasma strains affecting almond trees globally.

Introduction

Fruit species play a critical role in maintaining the global biodiversity and are essential to all living beings. Research on temperate fruit crops necessitates the fusion of fundamental and practical elements of plant physiology, ecosystem and genetics (Eyduran et al., 2015; Životić et al., 2019). The almond (Prunus dulcis [Mill.] D. A. Webb.), which is widely cultivated and holds significant commercial value as a tree nut crop, originates from the Mediterranean region. The almond fruit comprises an outer hull and a hard shell enclosed in the seed, which is a drupe (Gomez et al., 2007). The mesocarp of the almond is leathery and dry, which distinguishes it from other Prunus species and it dehisces at maturity. The almond trees thrive best in a temperate etesian climate with mild winters, frost-free, rainless springs and warm summers for nut ripening (Wang et al., 2021). Almonds are known for their rich nutritional value, including oleic and linoleic fatty acids, protein, dietary fibre, necessary nutrients such as calcium, phosphorus and magnesium and vitamins which contribute significantly to healthy living. More than 50% of the world’s almonds production is contributed by California, which accounts for around 8% of the total worldwide production of 3,214,522 tonnes annually, while Spain comes in second place with an annual production of 202,339 tonnes (Kumar, Bhuj & Dhar, 2023). In India, almonds are one of the most significant nut crops in temperate region and were first introduced in Kashmir during the 16th century by Persian settlers. The consumption of almonds in India is projected to reach 97,000 MT in MY 2017/18, an increase of 10% from the previous year. Despite this increase, India’s almond production only accounts for 2.45% of the world’s almond production (Wani & Bhat, 2021).

Almond is susceptible to a range of diseases caused by fungi, bacteria, viruses and phytoplasma diseases. Phytoplasmas are unculturable, wall-less bacterial plant pathogens belong to class Mollicutes, are a significant threat to almond trees in many countries causing serious diseases such as yellows and decline (Bertaccini & Lee, 2018). They infect plants through homopteran insects, such as leafhoppers and planthoppers, which feed on the phloem sieve tube elements. Plants infected with phytoplasma exhibit a range of symptoms leading to the development of green leaf-like structures instead of flowers (known as virescence/phyllody), sterility of flowers, abnormal growth of axillary buds resulting in “witches’ broom” symptoms, abnormal elongation of internodes and generalized stunted growth (Bertaccini, 2007). Moreover, their genomes signify that phytoplasma has a small genome than their bacterial ancestors as they lack several pathways for synthesizing compounds necessary for survival and have limited metabolic pathways (Kube et al., 2012; Oshima, Maejima & Namba, 2013; Marcone, 2014). Phytoplasma are typically found residing in the phloem sieve tube elements of diseased plants and are transmitted from plant to plant by phloem-feeding homopteran insects, mainly leafhoppers (Cicadellidae) and planthoppers (Fulgoromorpha) and less frequently psyllids (Psyllidae) (Weintraub & Beanland, 2006). While phytoplasma can cause symptoms in their wild reservoirs, they can also infect economically important fruit crops worldwide (Bertaccini et al., 2014; Fiore et al., 2018) but limited information is available on phytoplasma occurrence on fruit crops in India (Rao, Rao & Kumar, 2020). Although phytoplasma DNA has been found in seeds from phytoplasma-infected plants, it remains unclear whether they are seed-borne pathogens. Additionally, phytoplasma cannot be transmitted mechanically (Contaldo et al., 2012; Contaldo et al., 2016) and instead is spread through the use of infected vegetative propagating material (Dickinson, Tuffen & Hodgetts, 2013). While phytoplasma diseases were originally classified as auxenic, which refers to their effect on plant host growth, it is now evident that their pathogenicity involves a variety of effector proteins that are secreted into the cytoplasm of the phloem sieve cells. These effector proteins have diverse effects not only on plant growth but also on other aspects of plant life. Although it is challenging to grow phytoplasma under in vitro conditions due to their obligate parasitic lifestyle, continued research is essential to understand and mitigate phytoplasmal diseases (MacLean et al., 2011; Anabestani et al., 2017; Kitazawa et al., 2017).

Accurate and sensitive detection is a prerequisite for the management and study of diseases associated with phytoplasma. Initially, detecting phytoplasma was challenging due to their low concentration in woody hosts and their erratic distribution within infected plants (Berges, Rott & Seemuller, 2007). Techniques such as electron microscopy, DAPI (DNA-specific dyes) and graft transmission were previously used, but they were unable to differentiate phytoplasma. Later, serological techniques were developed with the production of enriched antigens specific to phytoplasma, but their application was restricted due to challenges in producing antisera (Bertaccini & Duduk, 2009). Around 20 years ago, the field of phytoplasmology underwent a revolutionary change with the advent of DNA-based techniques, leading to the development of specific and precise detection methods, predominantly utilizing the polymerase chain reaction (PCR) assay, which enabled the differentiate, classification and characterization of phytoplasma based on their 16S ribosomal DNA (rDNA) sequence and restriction fragment length polymorphism (RFLP) analysis. As a result, more than 30 ‘Candidatus phytoplasma’ species have been officially recognized and around 20 major phylogenetic groups or subclades have been identified. Recently, computer-simulated RFLP analysis method has recently increased the number of 16Sr groups and subgroups to 31 and over 100, respectively. These advances in phytoplasmology have greatly enhanced our ability to detect and identify phytoplasma, thereby assisting in the exploration and management of diseases associated with phytoplasma (Davis et al., 2013).

Stone fruit trees are susceptible to a variety of phytoplasma strains that belong to at least 14 different ‘Candidatus Phytoplasma species’ were reported globally (Zirak, Bahar & Ahoonmanesh, 2009). Among phytoplasmal diseases in almond, almond witches’ broom, European stone fruit yellows, peach yellows and X-disease being the most prevalent around the world. In recent studies, several groups of phytoplasma such as ‘Candidatus Phytoplasma asteris’ (16SrB-I), ‘Candidatus Phytoplasma phoenicium’ (16SrIX-B) and ‘Candidatus Phytoplasma aurantifolia’ (16SrII-B) have been associated with almond trees worldwide (Zirak et al., 2021). In Lebanon and Iran, a new and devastating disease of almond (Prunus dulcis) trees showing almond witches’ broom or ‘almond brooming’ was associated with ‘Candidatus Phytoplasma phoenicium’(pigeon pea witches’ broom group; 16SrIX) (Choueiri et al., 2001; Verdin et al., 2003). However, there have been no previous reports of phytoplasma-associated diseases in almond trees in India where almond is commercially and economically important nut crop in temperate region. Therefore, detecting and identifying the pathogens that affect almond production is of utmost importance. To address this issue, a research investigation was carried out to ascertain the occurrence of phytoplasma in symptomatic and asymptomatic almond trees. Fluorescent microscopic studies were conducted and the 16S rRNA gene was amplified and sequenced to identify the pathogen. This study will provide valuable information for farmers and researchers, enabling them to develop effective strategies to manage the disease and maintain healthy almond trees.

Materials and Method

Sample collection

In the recent surveys during vernal (May to June) and autumnal (September to November) periods in 2020 to 2022, phytoplasma suspected symptoms of leaf yellowing, reddening, tattering and die-back were observed on almond from experimental farm of Fruit Science, Dr. YS Parmar University of Horticulture and Forestry, Nauni, Solan region. The symptomatic trees were noticed scattered throughout the orchards with no apparent pattern. Leaf samples from four symptomatic almond trees along with two asymptomatic trees from the respective orchards were collected. To determine, if the symptoms observed were associated with phytoplasma, leaf samples from both symptomatic and asymptomatic fruit trees were examined through fluorescent microscopic studies and further through nested PCR assays and frozen in liquid nitrogen until processed.

Fluorescent microscopy

Fluorescent microscopic detection of phytoplasma was conducted by using DNA-specific fluorochrome 4,6-diamidino-2-phenylindole (DAPI) by following Seemuller (1976) with little modification. Leaf midrib samples from both diseased and healthy plants were utilized to detect phytoplasma in phloem tissues. The leaves were cut into one cm lengths and submerged in sterile distilled water. To facilitate section cutting, the cut leaf samples were embedded in a small piece of potato tuber. Subsequently, the samples were immersed in a 5% glutaraldehyde solution and fixed for a period of 25–30 min. After removing them from the glutaraldehyde solution, the samples were rinsed with 0.1 M phosphate buffer (pH 6.9) for 5 min. The sample sections were then treated with the DNA-specific fluorochrome DAPI stain (4,6-diamidino-2-phenylindole) for a duration of 30–35 min. Once adequately stained, a cover slip was carefully placed over the sections and gently pressed with filter paper to remove any excess stain. These sections were then scrutinized using a high-efficiency epifluorescent microscope (EVOS Cell Imaging System) equipped with an excitation filter with a wavelength range of 300–400 nm and a barrier filter allowing transmitting light above 400 nm.

DNA extraction

After confirming phytoplasma presence in suspected almond samples through fluorescence microscopic studies, about 300 mg of leaf midribs were subjected to total nucleic acid extraction from both symptomatic and asymptomatic trees according to Zhang, Uyemoto & Kirkpartick (1998) as modified by Abou-Jawdah et al. (2002) with minor modification. This method involves sample disruption in liquid nitrogen using CTAB extraction buffer in pre-chilled mortar and pestle. All incubation was carried out for 30 min and the aqueous phase was carefully transferred to another centrifuge tube without disturbing the interphase. Subsequently, 2/3 equal volume of isopropanol or 7.5 M ammonium acetate was added to the aqueous phase, followed by an overnight incubation at −20 °C and nucleic acid pellet obtained from leaf samples was then suspended in TE buffer (500 µl) for further analysis. Furthermore, extracted DNA samples were quantified using PicoDrop (Eppendorf BioSpectrometer® fluorescence; Eppendorf, Hauppauge, NY, USA). The concentration of extracted DNA was also estimated in 0.8% agarose gel electrophoresis stained with ethidium bromide.

PCR assays

For phytoplasmas detection, PCR amplifications were carried out using universal primer pairs P1/P7 (Deng & Hiruki, 1991) which produce 1,784 base pair long 16S rDNA fragment in the first round subsequently by internal primers PD1/PD2 (Specific probes prepared at NBRI) which produce 1,600 base pair long 16S-23S rDNA fragment (Khan et al., 2013) for phytoplasma detection. For the first round of PCR using primer pairs P1/P7, the reaction conditions consisted of one cycle of 94 °C for 5 min, followed by 35 cycles of 94 °C for 60 s, optimal annealing temperature for 60 s and 72 °C for 2 min, followed by final extension at 72 °C for 5 min. The resulting PCR product was then diluted (1:20) and used as template for nested PCR using primer pairs PD1/PD2 under the same PCR conditions were performed: 94 °C for 5 min, 30 cycles of 94 °C for 60 s, optimal annealing temperature for 50 s, 72 °C for 1min, followed by 72 °C for 5min. To validate the PCR assay, Peach yellow leaf roll phytoplasma samples infected with a 16SrV (Elm yellows group) (Khan et al., 2013) and a healthy almond DNA sample were utilized as positive and negative control in PCR assays. The reaction mixture for each PCR assay in 25µl contained 10X PCR buffer, 25 mM MgCl2, template DNA of 6µl, 10 mM dNTP mix, 1 µl each of primer pairs (100 pmol) & 3U/µl Taq DNA polymerase. The PCR amplifications were performed in thermocycler (Proflex ™ PCR system). Following amplification, each PCR product was subjected to electrophoresis in 1% agarose gel in 1xTAE buffer & stained in ethidium bromide. The resulting PCR bands were examined under Gel Doc XR+ Gel Documentation System for analysis.

Sequencing and phylogenetic analyses

The PCR products that were amplified with nested primers PD1/PD2 were carefully selected and subsequently purified using HiYield™ Gel / PCR DNA Mini Kit (Real Biotech, Banqiao City, Taiwan). Once purified, these products were sent to Eurofins It Pvt. Ltd. (Bangalore, India) for sequencing. To ensure comprehensive coverage of the amplicons, sequencing was carried out in both directions using PD1/PD2 nested primers as previously used for amplification through using Sanger sequencing. The sequences that were obtained from the chromatograms were aligned and assembled using Bio Edit software developed by Hall (1999). For reference purposes, the representative strains’ edited sequences were submitted in the GenBank data library maintained by NCBI in Bethesda, MD, USA and accession numbers were also received. Phylogenetic trees were constructed using the MEGA 11 developed by Tamura, Stecher & Kumar (2021). Specifically, a maximum parsimony tree was constructed of nearly identical length of 16S rDNA genes and submitted in the NCBI. As an outgroup, an Acheloplasma brassicae strain was used to root the tree with 500 replications of bootstrap analyses were performed.

Results

Almond phytoplasma disease symptoms

Throughout the period spanning from 2020 to 2022, surveys were conducted during both vernal and autumnal seasons revealed that almond trees in the Nauni region of district Solan (Latitude –30.91°N; Longitude –77.10°E) began exhibiting symptoms of the aster yellows disease in the mid-spring specifically during the month of June. The primary symptoms observed in affected almond trees included chlorosis, typical longitudinal, upward rolling (Fig. 1), reddening (Fig. 2), tattering and irregular leaf scorching (Fig. 3). Initially, these symptoms were thought to be associated with phytoplasma infection and were thus subjected to sampling. Furthermore, the branches were bearing small-sized fruits which often dropped prematurely ultimately resulting in decline, dieback and death of plants (Fig. 4) were observed in 40% of trees in almond orchards near the Dr. YS Parmar University of Horticulture and Forestry, Nauni, Himachal Pradesh, India.

Figure 1 Almond leaves showing chlorotic spots and upward rolling.

Almond trees showing main symptoms including chlorosis, typical longitudinal, upward rolling, reddening, tattering and irregular leaf scorching.

Figure 2 Distinctive symptoms.

Almond leaves exhibiting distinctive midvein and veinlets along with reddening.

Figure 3 Typical leaf symptoms.

Almond leaves exhibiting water soaked blotches along with red spotting and tattering.

Figure 4 Typical symptoms on trees.

Diseased almond trees exhibiting decline and die back symptoms.

Fluorescent microscopy

Fluorescent microscopy is a fast and effective histochemical test for the diagnosis of phytoplasma disease microscopically in the phloem of woody plants. The DAPI (4, 6-diamino-2-phenylindole HCL) test is capable of detecting the pathogen even when it is present in very low concentration in the diseased almond trees. In this study, the EVOS® FL Cell Imaging System was used to conduct fluorescent microscopic studies on sections of phloem tissues prepared from the shoot leaf petiole and mid rib of diseased almond leaves. The DAPI test results revealed distinct blue fluorescence in the sieve tube elements of the diseased almond tree samples, indicating the occurrence of phytoplasma (Figs. 5A, 5B, 5C). However, the blue fluorescence of phytoplasma bodies was not observed in the sample prepared from healthy trees. These findings suggest that the DAPI test is a reliable tool for detecting phytoplasma disease in almond trees even at low concentrations. The use of fluorescent microscopy, coupled with the DAPI stain, can aid in the early detection and management of phytoplasma disease in almond orchards.

Figure 5 Fluorescent microscopic studies.

(A) Healthy control. (B) Section of shoot leaf petiole showing fluorescent phytoplasma bodies. (C) Fluorescence in the sieve tube elements of leaf midrib.

Detection of phytoplasma by nested PCR assay

Nested PCR is a technique that utilizes a set of primers to amplify a specific DNA sequence of interest in two rounds of amplification. In this study, the primer pair P1/P7 was used in the first round to amplify a target sequence, followed by the use of the internal primer pair PD1/PD2 to amplify a smaller, nested sequence within the original product. This technique was applied to samples of four symptomatic almond samples and a positive control from the Elm yellows group (16SrV). The results revealed that the nested PCR successfully amplified a product of 212 bp from all of these samples viz. symptomatic almond DNA templates as well as from positive control (Elm yellows group; 16SrV) indicating the presence of the target sequence. However, neither direct nor nested PCR tests utilising DNA from healthy almond trees revealed any DNA bands (Fig. 6). These findings highlight the efficacy of nested PCR for detecting specific DNA sequences and its potential applications in various fields, including disease diagnosis and genetic research.

Figure 6 Gel electrophoresis images.

Agarose gel (0.8%) showing amplicons (∼212 bp) obtained from infected almond samples and peach samples. M: 100 bp DNA Ladder (Genei), Lane 1–4: Prunus dulcis (Diseased), Lane 5–6: Prunus persica (diseased; positive control), HC: Prunus dulcis (Healthy control).

Identification of phytoplasma infecting almond through phylogenetic analysis

For the identification and characterization of almond phytoplasma, four almond samples with a 212 bp nested PCR product were chosen for sequencing. The criteria for selecting the phytoplasma strains were based on the target tree, disease symptoms and the sample collection region. The nucleotide sequences of 212 bp amplified from the affected almond trees in the Nauni region were submitted to the NCBI GenBank database under the accession numbers OQ200126 (UHF1, Nauni). BLASTN searches of sequences showed that 16S rDNA nucleotide sequences of ‘Prunus dulcis’ aster yellows phytoplasma UHF1 strain shared 94% similarity with ‘Prunus dulcis’ phytoplasma PAEs3, ‘Prunus dulcis’ phytoplasma PAE28, ‘Prunus dulcis’ phytoplasma PASH1, ‘Prunus dulcis phytoplasma PAT6 and ‘Prunus avium’ phytoplasma PCHE1 (Zirak, Bahar & Ahoonmanesh, 2009; Zirak et al., 2021), respectively (Fig. 7).

Figure 7 Phylogenetic analysis.

Phylogenetic tree was constructed by the maximum parsimony tree method of nearly identical length of 16S rRNA gene from almond phytoplasma (UHF1) and 45 representative phytoplasmas deposited in NCBI. Bootstrapping was performed to support the branches in 500 replications. An Acheloplasma laidlawii (accession number M23932) strain was used as an outgroup in tree construction.

Discussion

Indigenous traditional occupations such as agriculture, animal husbandry and horticulture remain the foundation of Himachal Pradesh’s economy and have not been affected by modern industrial development. The production of almonds in the state has been increasing annually, resulting in a growing share of national production. Several pathogens infect almonds, with ’Candidatus Phytoplasmas’ members being a serious emerging pathogen causing severe yield losses. In the present investigation, we observed almond trees in Nauni region exhibited most prominent symptoms of aster yellows disease including leaf yellowing, reddening, red spotting, tattering and decline. Despite collecting both symptomatic and asymptomatic almond trees during the spring and autumn seasons, only symptomatic samples collected during the autumn season tested positive for phytoplasma using nested PCR assays. Previous studies have indicated that phytoplasma is eliminated from the above-ground portions of trees in cold temperature during the winter months, but the root system provide a safe haven for their survival allowing them to recolonize the stem and branches in the spring. In addition, the inactivity of phloem tissues in the aerial parts of stone fruit trees during the winter is responsible for the eradication of phytoplasma in the stem (Schaper & Seemuller, 1982; Seemüller, Schaper & Zimbelmann, 1984). Moreover, the movement of phytoplasma from the aged to the fresh phloem would exhibit variation from year to year, depending on the prevailing weather conditions that could influence the count of operational old sieve tubes. Detecting phytoplasma on stone fruit trees during the end of spring is a challenging risk (Jarauasch, Lansac & Dosba, 1999). It has been proposed that the difficulty in detecting phytoplasma during late spring is closely associated with the duration required for phytoplasma to grow on stems, branches, and new leaves (Errera, Aguelo & Hormaza, 2002). To identify the presence of phytoplasma in suspected almond samples, fluorescent microscopy with DAPI stain was employed due to its high sensitivity. DAPI stain is known to specifically bind with DNA of the phytoplasma in infected plant tissues leading to distinctive fluorescence in the phloem tissues. The strong fluorescence observed confirmed the presence of phytoplasma which confirmed the disease’s phytoplasmal etiology. After extracting plant DNA, a nested PCR method was employed because to its high sensitivity for detecting almond phytoplasma. Our findings demonstrated that primer pair P1/P7 and PD1/PD2 were used in the first round and nested PCR, expected DNA fragments of nearly 212 bp were amplified in nearly 40% of symptomatic almond trees samples as well as positive control (Elm yellows group; 16SrV), respectively. Furthermore, asymptomatic almond trees did not yield any PCR products. The obtained nucleotide sequences of 212 bp amplified from the affected almond trees within Nauni region was submitted to NCBI Gene Bank database, under the accession numbers OQ200126 (UHF1, Nauni). The 16S rDNA sequence analysis of almond phytoplasma revealed that 16S rDNA nucleotide sequences of ‘Prunus dulcis’ aster yellows phytoplasma UHF1 strain exhibited maximum identity (94%) with ‘Prunus dulcis’ phytoplasma PAEs3, ‘Prunus dulcis’ phytoplasma PAE28, ‘Prunus dulcis’ phytoplasma PASH1, ‘Prunus dulcis phytoplasma PAT6 and ‘Prunus avium’ phytoplasma PCHE1 reported by Zirak et al. (2021). Furthermore, almond has been identified as a novel host for certain subgroups of phytoplasma from Himachal Pradesh. Through phylogenetic analysis, ‘Prunus dulcis’ aster yellows phytoplasma UHF1 strain was classified within the 16SrI phytoplasma group (aster yellows group) strains, with the closest similarity to the strains of 16SrI-B subgroup (Candidatus Phytoplasma asteris). Although ‘Candidatus Phytoplasma asteris’ (16SrI-B) was previously been detected in apricot causing decline disease in Siot (Jammu & Kashmir) (Rao, Rao & Kumar, 2020) and in peach causing peach phytoplasma in Meerut (Uttar Pradesh) (Singh, Rani & Kumar, 2014), but its association with almond in India has not been previously reported. Therefore, we suggest that this is the first documented report of the occurrence of phytoplasma related to members of aster yellows phytoplasma (16SrI-B) group in almonds.

Conclusion

The presence of ’Candidatus Phytoplasma asteris’ in various hosts, including trees, perennials, and annual plants, indicates the possible involvement of a common and efficient insect vector(s), which will almost certainly cause outbreaks in the near future. Stone fruit is asexually propagated, so cuttings used for planting can transmit the disease. Therefore, it is critical to index established stone fruit nurseries for phytoplasma disease-free mother stocks. To improve disease management strategies in almond and other related Prunus species, molecular identification and characterization can contribute to our understanding of the genetic diversification and epidemiological studies of 16SrI phytoplasma as well as aid in identifying potential vectors in India as well as worldwide.

Supplemental Information

Supplemental Information 1 Sequence retrieved from GenBank for phylogenetic analysis

Click here for additional data file.

We thank the Dr YS Parmar University of Horticulture and Forestry for providing access to infrastructure, electronic resources, and laboratory facilities.

Additional Information and Declarations

Competing Interests

Author Contributions

Data Availability

Ravinder Kumar is an Academic Editor for PeerJ.

Shivani Gupta conceived and designed the experiments, performed the experiments, prepared figures and/or tables, and approved the final draft.

Anil Handa conceived and designed the experiments, authored or reviewed drafts of the article, and approved the final draft.

Ajay Brakta performed the experiments, prepared figures and/or tables, and approved the final draft.

Gulshan Negi performed the experiments, authored or reviewed drafts of the article, and approved the final draft.

Rahul Kumar Tiwari analyzed the data, prepared figures and/or tables, and approved the final draft.

Milan Kumar Lal analyzed the data, authored or reviewed drafts of the article, and approved the final draft.

Ravinder Kumar analyzed the data, authored or reviewed drafts of the article, and approved the final draft.

The following information was supplied regarding data availability:

The ’Prunus dulcis’ aster yellows phytoplasma isolate UHF1 sequence is available at GenBank: OQ200126.

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
