# Peer review of "First report of ‘Candidatus Phytoplasma asteris’ associated with yellowing, scorching and decline of almond trees in India"

_PeerJ, doi:10.7717/peerj.15926_

## Round 0.1 · original submission · Major Revisions

Authors are advised to revise the manuscript as per suggestions of the reviewers.

Reviewer 1 ·

Basic reporting

Language improvement is required in introduction and discussion.

Figures are well explanatory

Experimental design

Research design is appropriate. Title of the article, aims and objectives and methodology of article correspond well.

Research question is well formed

Validity of the findings

This study has reported for the first time the association of Candidatus Phytoplasma asteris’ infection in Almond Trees in India. That is why the paper has merits and the authors have performed molecular detection and fluorescent microscopic studies. The manuscript should be exhaustively revised to avoid any conceptual, typographical and grammatical errors.

Additional comments

1. The title is not justifying the study please mention it as the first report and remove characterization as single gene has been used for identification.
2. Modify line 11-12 : The almond, a commercially important tree nut crop worldwide, is native to the Mediterranean region.
3. Correct line 15: consistently affected?
4. Recheck the phylogenetic identity results and incorporate modifications if observed any variation.
5. Line 40-41: such common phrases are not required in a scientific study.
6. Correct lines 57-58
7. Line 115: rectify the error.
8. Line 126-127 please add further detail of the location
9. What methods have been previously used in the detection of phytoplasmas in almond trees? Those references should be included in this manuscript.
10. it’s a general suggestion to the authors to observe vector populations in the vicinity of those regions to establish phytoplasma-vector relation studies.
11. Please correct the grammar in sections such as introduction and discussion.

·

Basic reporting

Authors of the manuscript “Detection and molecular characterization of Candidatus Phytoplasma asteris associated with almond trees in India” provided first host report of a Candidatus Phytoplasma asteris infecting almond in India, expanding the knowledge of the diversity and distribution of phytoplasma strains affecting almond trees globally.
The manuscript is well-written in an engaging and lively style. The level is appropriate to the journal’s readership. The subject is very important, it’s currently something of a “hot topic” in almond growing areas, and this study makes significant contributions. This article may be accepted for the possible publication after correction of a few typos, grammatical mistakes and other minor issues.
Keywords must be different from title to enhance search ability and findability. Select words that describes the highlight/novelty of the research. Arrange these alphabetically.
L33: Italicize botanical names.

Experimental design

L17: Why surveys were conducted during May to 17 September?
What is null hypothesis of this study?
Give further detail about the status of almond-associated phytoplasmas in India and rest of the world.
L122: In the recent surveys during vernal (May to June) and autumnal (September to November)…? When surveys were conducted?
L132: Please maintain uniformity while writing subheadings.
Did you follow Kochs postulates during the study?
Give geographical locations of surveyed areas.
L138: The sample bits (about 2mm in size) will be stored at 4ºC in 0.1 M?
L140: …. thickness will be prepared and these section will be transferred to clean glassslide?
L142: The sections will be covered by cover slip and blotted firmly with filter paper ..?
What positive control was used during this study/ Please explain

Validity of the findings

Please make corrections in the phytoplasma species groupings mentioned in results section.

Additional comments

Please highlight the screened isolates used during this study in the phylogenetic tree.
Please write the complete name of an organism or term before writing its abbreviation. Afterwards, no need to write the complete term in a section.
Confirm either it is first study in India?

---

## Round 0.2 · accepted · Accept

Authors have revised manuscript as per suggestion. Therefore, It is recommended that manuscript to be accepted for publication.

The Section Editor noted:

"fluoresced" should be replaced with "fluorescent" (4 instances). line 26 "Aster" should not be capitalized

Reviewer 1 ·

Basic reporting

it seems to be connected now with respect to title, methodology and reported results

Experimental design

Well defined, Relevant, good standards

Validity of the findings

Satisfied with the proposed findings and rigorous discussion on it

·

Basic reporting

Authors have improved their manuscript. Now, it may be accepted for the possible publication.

Experimental design

Acceptable

Validity of the findings

Valid